# Design and Implementation of a Camera-Based Tracking System for MAV Using Deep Learning Algorithms

**Stefan Hensel** [1] , **Marin B. Marinov** [2,*] and **Raphael Panter** [1]

1 Department for Electrical Engineering, University of Applied Sciences Offenburg, 77652 Offenburg, Germany; stefan.hensel@hs-offenburg.de (S.H.)

2 Department of Electronics, Technical University of Sofia, 1756 Sofia, Bulgaria

\* Correspondence: mbm@tu-sofia.bg

**Abstract:** In recent years, the advancement of micro-aerial vehicles has been rapid, leading to their widespread utilization across various domains due to their adaptability and efficiency. This research paper focuses on the development of a camera-based tracking system specifically designed for low-cost drones. The primary objective of this study is to build up a system capable of detecting objects and locating them on a map in real time. Detection and positioning are achieved solely through the utilization of the drone's camera and sensors. To accomplish this goal, several deep learning algorithms are assessed and adopted because of their suitability with the system. Object detection is based upon a single-shot detector architecture chosen for maximum computation speed, and the tracking is based upon the combination of deep neural-network-based features combined with an efficient sorting strategy. Subsequently, the developed system is evaluated using diverse metrics to determine its performance for detection and tracking. To further validate the approach, the system is employed in the real world to show its possible deployment. For this, two distinct scenarios were chosen to adjust the algorithms and system setup: a search and rescue scenario with user interaction and precise geolocalization of missing objects, and a livestock control scenario, showing the capability of surveying individual members and keeping track of number and area. The results demonstrate that the system is capable of operating in real time, and the evaluation verifies that the implemented system enables precise and reliable determination of detected object positions. The ablation studies prove that object identification through small variations in phenotypes is feasible with our approach.

**Keywords:** convolutional neural networks; deep learning algorithms; MAV; object identification; tracking systems





## 1. Introduction

Micro-aerial vehicles (MAVs), especially in the form of quadcopter systems, have shown rapid technical development combined with a decrease in pricing in recent years.

By enhancing drone safety systems, accidents can be increasingly avoided and the risk of damage can be minimized. With the great improvement in camera technology, drones can capture sharp images and high-definition videos. The ability to fly autonomously without human intervention is enabling the use of drones in increasingly complex environments and scenarios, such as precision agriculture [1], surveillance for military and industrial applications, or logistics [2]. They are employed in long-range medical supply or postal delivery [3].

Significant progress has also been made in the field of deep learning over the last decade. As a result, it has become a significant tool in data analysis and pattern recognition. Especially in the field of computer vision, deep learning has been able to address a wide range of problems in image processing and object recognition [4,5]. Using convolutional neural networks, objects in images can be recognized and detected. If the detector is extended with multi-object tracking, these objects can additionally be tracked and analyzed within a scene [6].

Deep learning algorithms can process and analyze video data from drones in real time. The combination of these two technologies offers great potential to improve the performance and expand the application areas of drones.

However, these tasks are challenging due to the high mobility, limited computation, and varying viewpoints of MAVs.

There are three main research topics in DL-based object detection and tracking from MAVs: object detection from the image, object detection from the video, and object tracking from the video. Object detection from the image aims to locate and classify objects in a single image, while object detection from the video aims to do the same in a video sequence. Object tracking from the video aims to track the location and identity of objects across frames.

Some of the recent advances and challenges in these topics are summarized as follows:

- Object detection from the image: DL-based methods have achieved state-of-the-art performance on various UAV datasets, such as UAVDT, VisDrone, and DOTA. However, there are still some issues, such as small object detection, occlusion handling, and domain adaptation. Some of the representative methods are Faster R-CNN, YOLO, and SSD, which use different network architectures and loss functions to balance accuracy and speed [7,8].
- Object detection from the video: DL-based methods have improved the performance of object detection from the video by exploiting temporal information, such as optical flow, motion vectors, and recurrent neural networks. However, there are still some challenges, such as video compression artifacts, motion blur, and background clutter. Some of the representative methods are FGFA, DFF, and STSN, which use different ways to aggregate features from multiple frames and enhance temporal consistency [9,10].
- Object tracking from the video: DL-based methods have achieved remarkable results on various UAV tracking benchmarks, such as UAV123, UAVDT, and VisDrone. However, there are still some difficulties, such as scale variation, illumination change, and occlusion. Some of the representative methods are SiamFC, SiamRPN, and ATOM, which use different strategies to learn a similarity measure between the target and the candidates and optimize the tracking performance.

In conclusion, DL-based methods have shown great potential and progress in object detection and tracking from MAVs, but there are still many open problems and directions for future research [11–13].

So far, commercial applications have been restricted to high-end models with high pricing [14]. In this contribution, we propose to employ a low-cost quadcopter system and combine it with deep-learning-based methods for object detection and object tracking. The algorithms are divided between the drone and base station, which takes the role of an operation center (OC) for several scenarios. We deliberately choose two examples that we think are viable as real-world applications. In the first, a classical search and rescue scenario is assumed, and the drone is sent to a certain area, defined by GPS coordinates and satellite imagery. Autonomous path planning and real-time feedback of possible victims are vital in this scenario. Additionally, the drone can come back to OC and afterward take off to hover above a manually selectable victim. This allows for an efficient search strategy and the camera-enabled location narrows the position to submeter accuracy.

The second scenario we describe has its application in husbandry and livestock surveillance: the drone is capable of flying in a GPS-defined area and counting possible animals. The challenge in this case is the movement of possible animals and the need to distinguish each one to prevent double counting or missing out on an exemplar. Therefore, we propose to use deep-learning-based tracking algorithms that are engineered and adopted to track individuals over a certain period.

The rest of the paper is organized to describe the design and implementation of a system for the tracking of objects or people with the help of a quadcopter. We present the related work, its adoptions, and our developments for the described scenarios.

We investigate and evaluate the detection of multiple objects correctly in real time and track them over a longer period. A suitable interface for controlling the quadcopter and delimiting the flight area is presented. To ensure communication with the drone, middleware was created.

Object detection based on convolutional networks is explained and an adequate detector is selected after an evaluation. In addition, several state-of-the-art tracking algorithms are presented and assessed for their suitability in the presented scenarios. The GPS positions of the detected objects are to be determined and recalculated in a local coordinate frame concerning the OC and drone. The middleware must be able to announce the information of the calculated GPS positions at runtime and communicate orders from the operator to the drone.

To be able to cope with broad applications, we present an evaluation method to determine the minimum quantity of needed data to train an object detector. For the case of object tracking, a thorough investigation of the minimal difference needed between two objects is given to better understand which kinds of animals can be distinguished from each other.

## 2. Materials and Methods

### 2.1. Detection of Objects

Object detection is a computer vision technique [15]. This technique can be used to identify and locate objects within an image or video. The detected objects are marked with a bounding box. This allows users to determine where the objects are located in the frame. The most important measurement parameters are accuracy and speed. The detection of objects provides a basis for further computer vision tasks, especially instance segmentation, image recognition, and tracking of objects [16].

#### 2.1.1. Strategies

Besides the algorithms of the "traditional object detection period", such as the Vio-la-Jones detector [17] or the HOG detector [18], there are algorithms in the "deep-learning based detection period" which are based on two strategies [16].

*Two-stage detectors*: Two-stage detectors divide the localization and classification of objects into two stages. In the first stage, suggestions for regions of interest (RoIs) in the image are proposed [19]. In the second stage, the features of each proposed bounding box are extracted and used to classify them subsequently. By splitting into two phases, this type of detector has high localization and object detection accuracy. Well-known representatives of this two-stage strategy include Fast-R-CNN [20], Faster-R-CNN, and SPP-Net [21].

*Single-stage detectors*: Single-stage detectors combine the localization and classification of objects in one step [19]. Thus, they are time efficient and can be used in real-time applications. For this reason, this work focuses on detectors with a one-step strategy. Well-known representatives of the one-step strategy are SSD and You Only Look Once (YOLO) [22].

#### 2.1.2. YOLO

The convolutional network YOLO presents object recognition as a single regression problem. This method leads directly from the image pixels to the bounding box coordinates and the probabilities of the classes. With YOLO's system, the image only needs to be viewed once to identify which objects are present in it and where they are located. This is simple and straightforward. The system resizes the input image, runs it through a convolutional network, and evaluates the resulting detections against a threshold. The convolutional network can simultaneously predict multiple bounding boxes and their detection probabilities for each class [22]. YOLO is trained in complete images. It also directly optimizes recognition performance. This unified model has several advantages over conventional methods.

YOLOv5: There is a large number of different implementations of YOLO. YOLOv5 is used in this work, but other implementations include YOLOv3, YOLOv4, and YOLOv6, among others.

The network structure of YOLOv5 is divided into three parts [23]. As a backbone, YOLOv5 uses the CSPDarknet, and as a neck, PANet was adopted. By default, the network is passed an input image with dimensions $640 \times 640 \times 3$. The CBL module consists of a convolution, a batch normalization, and an activation function based on ReLU [24].

### 2.2. Multi-Objekt-Tracking

MOT plays an important role in the field of computer vision. The task of MOT is essentially to locate multiple objects, preserve the identities of the objects, and determine the individual trajectories of each object over the length of a video [25].

Processing Method

MOT can be divided into two types of processing. The difference between online tracking and offline tracking is whether observations from future frames are used when processing the current frame.

*Online tracking*: In online tracking, the video sequence is processed step-by-step. This allows only the current detections to be viewed. The existing trajectories are gradually augmented with the current observations. Online tracking algorithms can be used in real-time applications, unlike offline tracking methods. However, they are less robust in comparison, since only a few data of observations are available [25].

*Offline tracking*: An offline tracking algorithm processes the video sequence images in batches. In the offline tracker, all detections can be used. In this process, the tracker can theoretically determine a globally optimal solution. Offline tracking algorithms have a delay in outputting the final results.

For the implementation of the work, a tracking algorithm capable of tracking objects in real time is needed. One such online tracking algorithm for real-time applications is Deep Simple Online and Realtime Tracking (DeepSORT).

### 2.3. DeepSORT

DeepSORT is a powerful and fast algorithm [26]. This makes DeepSORT a leading algorithm for tracking objects. DeepSORT represents a stable baseline model for tracking algorithms as a result of its effectiveness, extensibility, and simplicity [27].

DeepSORT is an improvement of the SORT algorithm [28]. DeepSORT additionally pulls in information about the appearance of the detections to improve the association of existing objects.

### 2.3.1. SORT

The tracking methodology of the SORT algorithm mainly consists of the detection of objects, the transfer of object states to future frames, the association of current detection with the already existing objects, and the management of the actively tracked objects [29].

*Detection*: The Faster Region CNN (FrRCNN) framework is used for detection. This is a two-stage detector. The detector can be exchanged arbitrarily.

*Estimation Model*: The Estimation Model is used to describe the object model. In it, the representation and the model to transfer tracked objects to the next frame are defined. The displacement of an object between frames is approximated using a Kalman filter [30] with a linear constant velocity model. The state of each object is modeled as follows:

$$x = \begin{bmatrix} u, & v, & s, r, \dot{u}, & \dot{v}, & \dot{s} \end{bmatrix}^{T}. \tag{1}$$

Here, $u$ and $v$ represent the pixel coordinates of the center of the object, $s$ the scale, and $r$ the aspect ratio. If a detection can be associated with an already existing object, the state of the object is updated. The bounding box properties of the detection are used

for this purpose. The velocity components are optimally solved using a Kalman filter. If no detection can be associated with an object, its state is predicted using only the linear velocity model without correction.

Association: A cost matrix is created to associate detections and tracked objects [29]. In the matrix, the IoU distances between each detection and all predicted positions of the objects present are calculated. The mapping is solved using the Hungarian method. In addition, a threshold is used for the value of the IoU. If the value is lower than this, the mapping is not applied.

Creation and deletion of tracks: If a detection whose IoU value is above the threshold cannot be assigned, a new track is assumed. This is initialized with the values of the bounding box of the detection. To finally add the target object as a track, further detections must be able to be associated with it. Tracked objects are deleted when they are no longer detected for a certain number of frames [29].

### 2.3.2. SORT with a Deep Association Metric

DeepSORT uses a conventional single-tracking method with recursive Kalman filters and frame-by-frame data association [31]. The following section presents the core components of the DeepSORT algorithm. DeepSORT uses an eight-dimensional space to model the states of the tracks:

$$x = \left[ u, \ v, \ \gamma, h, \dot{u}, \ \dot{v}, \dot{\gamma}, \dot{h} \right]^{T}. \tag{2}$$

Again, $u$ and $v$ represent the pixel coordinates of the center of the object's bounding box. $\gamma$ describes the aspect ratio and $h$ the height. A standard Kalman filter with constant motion velocity and a linear observation model are used. The bounding box values ($u, \ v, \ \gamma, h$) are used to observe the object state.

For each track, the number of frames since the last successful assignment is counted. During Kalman filter prediction, this count is incremented and reset to zero if the track can be associated with detection. If a track cannot be associated for a predefined time, the object is assumed to have left the scene and been removed from the list of tracks. If a detection cannot be associated with an existing track, it is initialized as a new track. New tracks are marked as tentative. In this state, a successful association with a detection must occur at each subsequent time step. If no further association takes place in the first three frames, the track is deleted.

*Matching Problem*: DeepSORT also uses the Hungarian method [32] to solve the matching problem between predicted Kalman states and new detections. The Hungarian method is used to minimize the total cost of matches. In addition, a matching matrix is created that contains the labels of the successfully matched as well as the unsuccessfully matched targets.

In the statement of the problem, motion information and appearance features of the detections are integrated by combining two appropriate metrics [31].

To incorporate motion information, the (squared) Mahalanobis distance between predicted Kalman states and new detections is used. The Mahalanobis distance is an appropriate measure when motion uncertainty is low. The distance gives the distance between points in a multi-dimensional vector space in standard deviations. In this case, the Mahalanobis distance measures how many standard deviations away the detection is from the average track position. This accounts for the uncertainty in the state estimate. Furthermore, using this distance, a threshold can be formed at a 95% confidence interval. This makes it possible to rule out improbable associations.

Camera movements that are not considered can lead to rapid shifts in the image plane. This makes the Mahalanobis distance a rather inadequate metric for tracking by occlusion. Therefore, the cosine distance is integrated as a second metric in the assignment problem. For this purpose, an appearance descriptor is computed for each detection. In addition, the last 100 of these descriptors are stored for each track. To compute the appearance descriptors, a pre-trained CNN is used and a threshold is set.

In combination, both metrics complement each other as they cover different aspects of the assignment problem. While the Mahalanobis distance results are particularly useful for short-term predictions, the cosine distance information provides a way to recover tracks after long-term occlusions. For this reason, both metrics are combined in a weighted sum. The association is considered admissible if it is above the specified thresholds of both metrics.

*Matching Cascade*: Instead of solving a global assignment problem, a cascade is introduced that solves a set of subproblems [33]. It is designed to give preference to more frequently seen objects. This is because if an object is occluded for a longer period, the Kalman filter becomes more uncertain about its predictions of the object's position. As a result, the association metric increases the distance between detection and tracking. Then, when two tracks compete for detection, the Mahalanobis distance favors greater uncertainty. This is an undesirable behavior and leads to increased track fragmentation.

The DeepSORT algorithm follows a so-called matching cascade [31], which results in the following processing sequence:

1. Calculate the Mahalanobis distance and the cosine distance using the new detections.
2. Create the cost matrix using the weighted sum of both metrics, considering the thresholds set.
3. Iterate over tracks that could not yet be assigned to a detection.

    (a) Linear mapping of these tracks with non-matching detections using the created cost matrix and the Hungarian method.
    (b) Update the matching and non-matching detections.

4. Return the updated detections.

In the last phase of the *Matching Cascade*, the IoU association proposed by SORT is performed for the unconfirmed and unmatched tracks. This helps to account for sudden changes in the feature vector of detection due to partial occlusion. It also increases robustness to incorrect initialization.

*Deep-Appearance Descriptor*: Successful application of the DeepSORT method requires a well-discriminating feature extractor. This must first be trained offline before the actual online tracking application. For this purpose, CNN is used in DeepSORT. This was trained on a large person recognition dataset. The dataset includes over 1.1 million images of 1261 pedestrians. CNN is a wide residual network (WRN). An implementation of the DeepSORT algorithm and an example of the CNN can be found in [33].

For clarity, the basic flow of DeepSORT is explained briefly as follows: First, new detections are passed directly into the matching cascade [34]. If the detections can be matched to tracks, the tracking bounding box is updated using the Kalman filter. Detections and tracks that could not be matched or were not confirmed after the Kalman filter prediction are passed for the IoU association. After the IoU association, unassigned detections are included as new tracks. Tracks that remain unassociated are deleted or processed with the confirmed tracks, depending on their status and age. Confirmed tracks are prepared by the Kalman filter prediction for the processing of the next frame and made available to the matching cascade.

### 2.4. Parrot ANAFI

The *Parrot ANAFI* was specified as the drone for this work. The quadcopter was developed and is distributed by the Parrot company. The ANAFI is characterized as a lightweight and portable flying device, which makes it versatile in various applications. The drone is equipped with a 4K camera and is consequently particularly suitable for high-resolution aerial photography. In addition, the drone is addressable via a programming interface. The interface allows developers to access the functions and control options of the drone. This allows them to create their applications. Communication with the drone takes place via Wi-Fi [35].

## 3. Implementation

### 3.1. System Structure and Communication

In the first step of the implementation, the underlying system architecture was defined. The only criterion here was the modular structure of the components. The system and the individual modules should be easily expandable and interchangeable. The system architecture of the work is based on the client–server model. The server takes the role of a middleware.

There must be constant communication between the server and the drone. The communication is realized via a Wi-Fi network, which is established by the drone. The device running the server can connect to this network. Figure 1 shows an overview of the communication between the client–server and the drone. Communication via Wi-Fi is particularly suitable since almost all end devices are now able to use this technology. This means that the drone can be controlled by many different devices. The drone is controlled by the *Olympe* SDK. To send commands to the drone, a connection message must be sent from the server after connecting to the Wi-Fi network. If this is accepted by the drone, no other device can send commands.

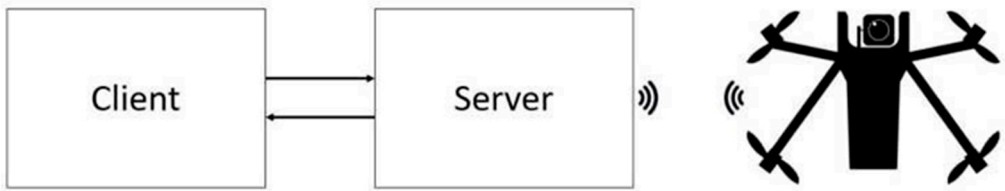

**Figure 1.** Communication within the system.

The communication between the client and server mainly takes place via a REST interface. All essential commands for controlling the drone are sent from the client to the server via this interface. These include, for example, starting and stopping the drone, calculating the flight data, and switching the video stream on and off. After successful execution, the server sends an acknowledgment to the client. In addition, bidirectional communication via web sockets is provided. This allows additional asynchronous messages to be exchanged. This communication is used to announce the positions of the detected objects. Figure 2 shows the timing of the communication between the client and server during a complete program run. The order of the interface calls is crucial. Changing the order is not allowed in this case and would lead to an abortion of the communication. The run-through in the figure always assumes the successful execution of the commands. The handling of errors is not considered in this example. First, the connection to the drone is established. At the same time, the client connects to the web socket of the server. Subsequently, the data can be transmitted to the observation area.

In Section 3.2 the execution of the determination of the flight area and the calculation of the flight route are explained in more detail. The calculated flight data can be found in the server's feedback. After the calculation of the flight route, the drone can be started. It can be decided whether the live stream of the drone should be displayed. During the overflight of the drone, the locations of the detected objects are published via a communication channel of the web socket. After completion of the flight, an overview of the detected objects is also returned. In this are the classes and the number of detected objects corresponding to the respective class. In the last step, a selected target is approached.

*User interface*: The user interface was initially developed simultaneously with the middleware. For this reason, it is also based on the Python web framework *Flask* for the sake of simplicity. However, it can be executed as a stand-alone application or via a *Node.js* server with minimal adjustments. In addition, external libraries had to be used. These were integrated via a content delivery network (CDN). Figure 3 shows a mockup of how the user interface should look during the planning phase.

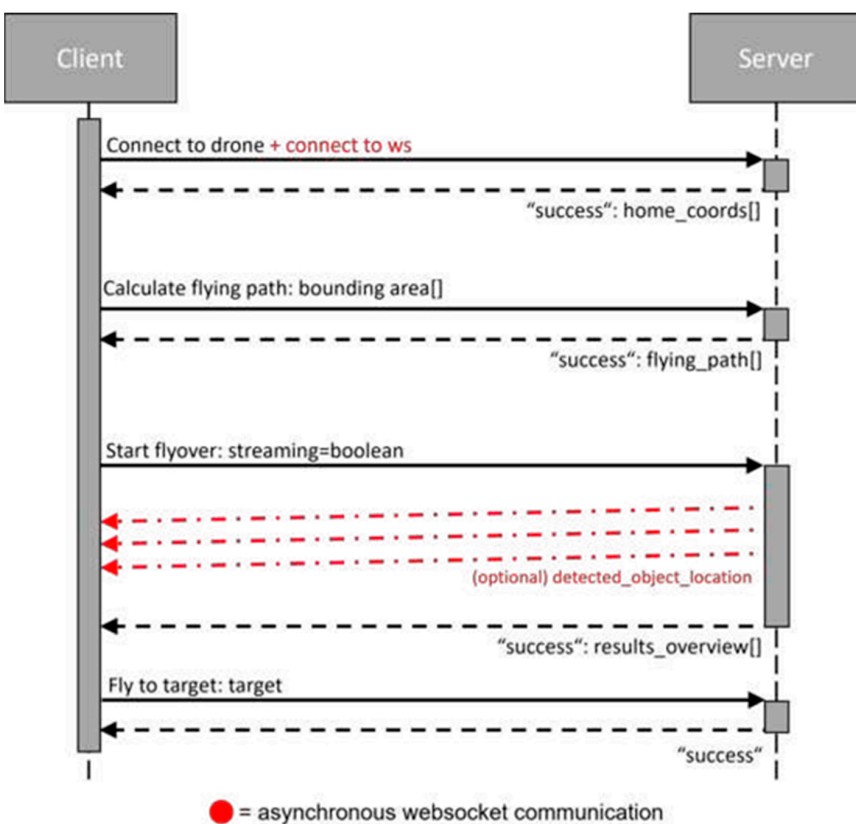

**Figure 2.** Timing of communication between client and server during a system scan.

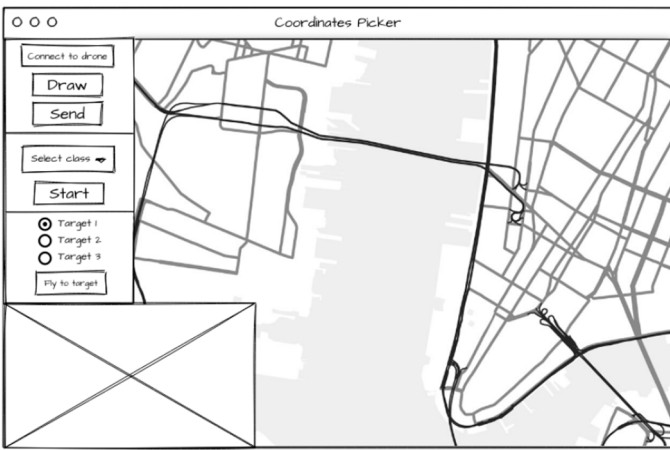

**Figure 3.** User interface mockup.

On the left side is a control panel. This is used to communicate with middleware. The map on the right side should be used to define a flight area. The placeholder at the lower left corner is reserved for the live stream of the camera image.

For the implementation of the user interface, a website was written. For this HTML was used for structuring the content, CSS for the design and layout of the page, and JavaScript for the functionality and interactivity. The website consists of two scenes. Figure 4 shows an overview of both scenes. Scene 1 is the landing page. There, a checkbox can be used to select whether the live stream of the drone should be displayed or not. The connection to the drone can be established via a button. The interface changes to the second scene when the button is clicked. The focus of the map shifts to the GPS coordinates of the drone.

The exact location of the drone is highlighted with a blue marker. With the help of the external library Leaflet and some extensions, the interactive map can be integrated. The user is enabled to interact freely and dynamically with the map. It is possible to zoom in and out and to view the environment. The appearance of the map is defined by the style tag. Many different tile servers are available for this purpose. For example, the appearance of OpenStreetMaps, Google Maps, or Google Maps Satellite can be adopted. In the context of the work, the Leaflet extension of Esri was merged. According to experience, this provides the most up-to-date maps. Furthermore, the extension allows an enlarged zoom. However, an API key is required to use the Esri extension. More detailed information on how to obtain the key can be found in [36].

**Scene 1**

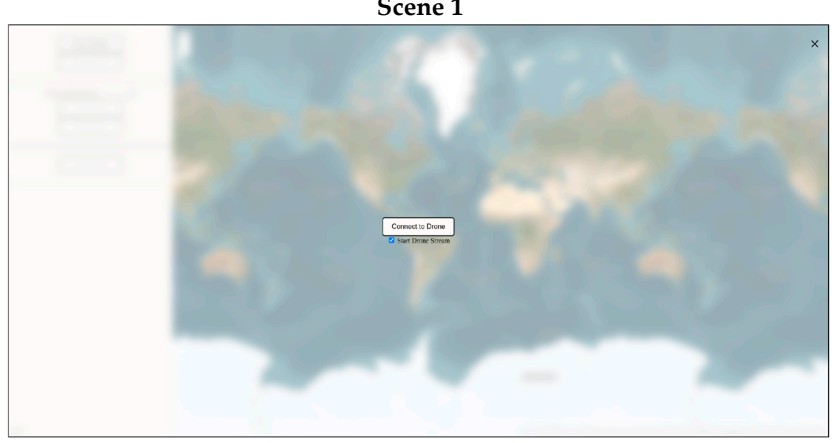

**Scene 2**

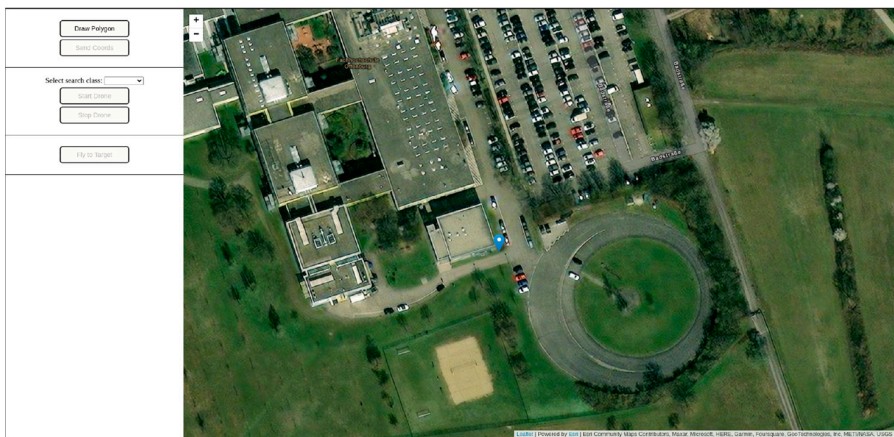

**Figure 4.** User interface implementation.

The *Leaflet Geoman* extension helps to draw and adjust polygons. To switch to the drawing mode, the "*Draw Polygon*" button must be pressed. This mode remains active until the polygon is completely closed. Then the polygon can be adjusted by moving the corner points. After completion of the polygon, an additional button appears. With this button, the polygon can be deleted. Only one polygon can exist and be drawn at a time.

The "*Send Coords*" button can be used to send the coordinates of the polygon's edge points to the server. In response, the client receives the planned flight route of the drone. This route is entered as a red path on the map using Leaflet Geoman. A drop-down list can be used to select the search classes provided. The *Start* and *Stop* buttons can be used to start and stop the search process, respectively. During the search process, the client receives information about the detected objects. This requires a bidirectional communication channel. Using the *SocketIO* library the client can connect to a web socket server and exchange messages asynchronously. The detected objects are successively entered with a marker on the map at runtime. Figure 5 shows an overview of the detected

objects that have been entered on the map. The red marker corresponds to the initial position of the drone. The blue markers represent the detected objects. A target can be selected via radio buttons. The color of the marker of the selected target changes accordingly during the selection. The "*Send to target*" button can be used to send the drone to the target.

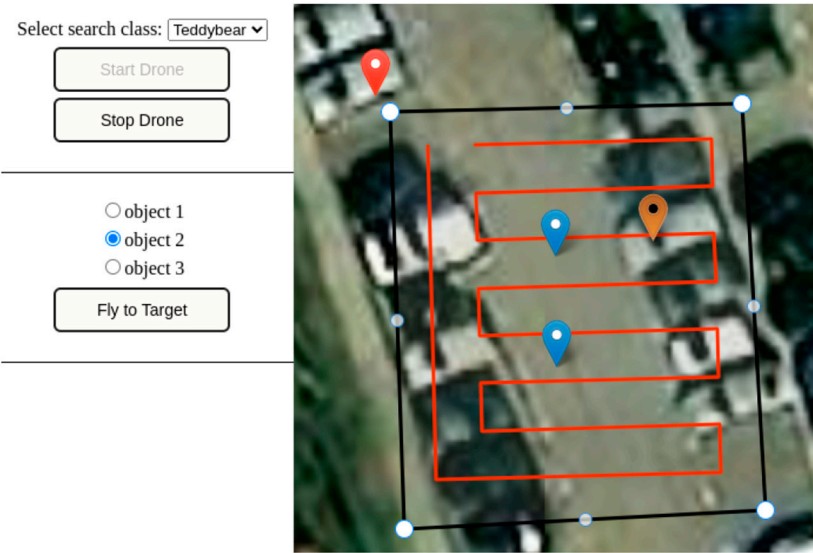

**Figure 5.** Overview of detected objects on the map. Red symbol indicates the starting position, blue ones are detected objects, the orange symbol is a manually selected object.

The buttons were configured to be enabled and disabled under certain conditions. This ensured that commands could only be sent in the order described in Section 3.1. Ajax calls were used to send the commands. To increase the efficiency of web development and facilitate this task, the *jQuery* library was included. This library already offers a complete implementation of the Ajax calls, so that the corresponding functions only have to be called.

### 3.2. Calculation of the Flight Route

At least three GPS coordinates are required to calculate the flight path of the drone. These are drawn in graphically by the user via the user interface and transmitted to the server. Figure 6 shows an example of the transmitted coordinates from the client. In this example, four coordinates were selected. These are highlighted with blue markers and span a polygon. The spanned polygon is accentuated in yellow and represents the desired observation area for the drone. The red marker shows the starting position of the drone. The Python library "Shapely" was integrated for the implementation. It is used for the manipulation and analysis of geometric objects. For this purpose, the library provides functions for creating and editing geometric data.

After the polygon has been formed, the outermost boundary values are read out. For this purpose, the minimum and maximum coordinate values of all points of the outer line of the polygon are determined. These coordinate values define the bounding box of the polygon. Figure 7 shows the polygon and the corresponding bounding box. Within this box, a rectangular grid is created with a uniform spacing. The centers of the grid cells correspond to a potential approach point of the drone. The size of the grid cells can be manually adjusted depending on the extent of the search area. Thus, the number of approach points can be adjusted. For this purpose, the coordinates had to be converted from the ellipsoidal coordinate system WGS84 [37] to a Cartesian coordinate system in the implementation. Due to its high accuracy, ETRS89 was determined as the target coordinate system. For the conversion, the library "pyproj" was used. This library forms a Python interface to the PROJ (version 3.6.1) software. PROJ performs the transformation of spatial coordinates between coordinate reference systems. By transforming the coordinate systems, the grid cells in meters could be created step-by-step. After creating the grid in meters, the individual approach points were

transformed back to the ellipsoidal coordinate system. The blue points in Figure 7 represent the possible approach points in the ellipsoidal coordinate system.

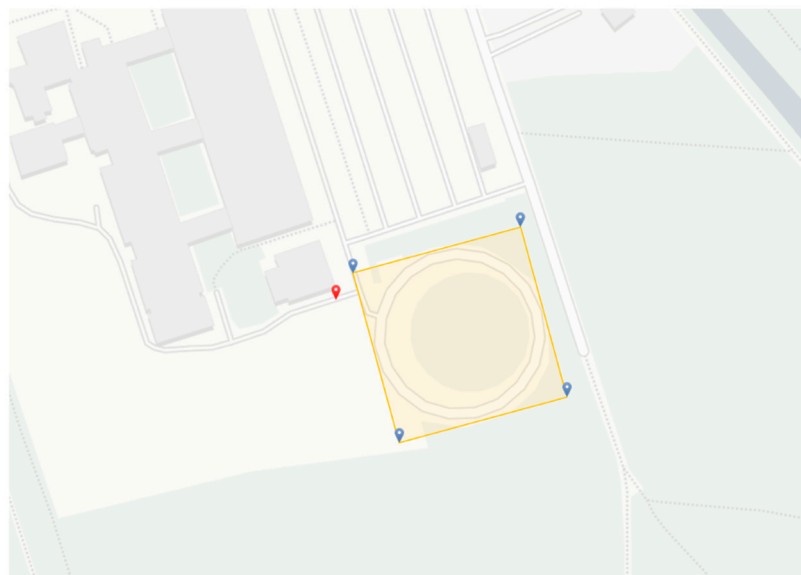

**Figure 6.** Definition of the flight area. Starting position of drone in red, blue markers indicate manually selected polygonal points.

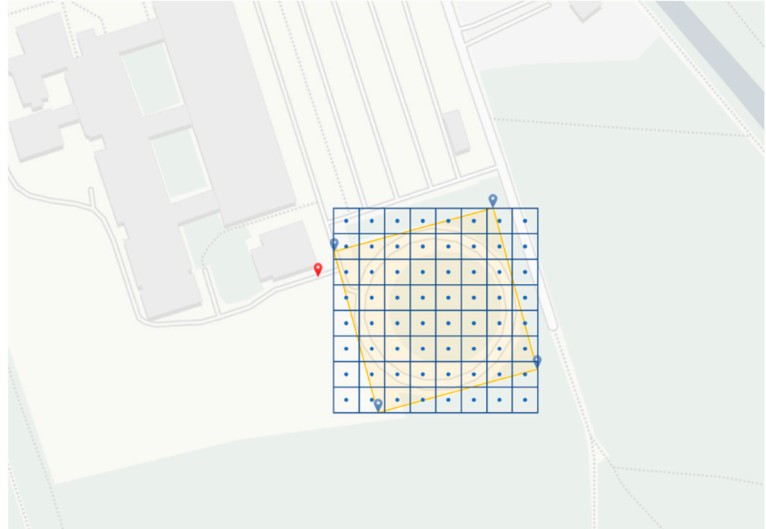

**Figure 7.** Building the grids within the bounding box. Yellow polygon in cartesian coordinates, the blue grid and center points are in ellipsoidal coordinates for GNSS (WGS84).

In the next step, the flight route points are divided into two classes. These classes determine whether the points are inside or outside the flight envelope. In Figure 8a, the points inside the flight range are colored in green and the points outside are colored in red. For this purpose, a function of the Shapely library could be used to determine the points that are inside the polygon. The points that lie outside the polygon were not considered further and were finally discarded. This resulted in the state shown in Figure 8b. If desired, it would also be possible to include the points that lie on the edge of the polygon. This would also cover possible exceptional cases that are close to the boundary of the search area. With the set of the remaining points, the flight route can be planned afterward.

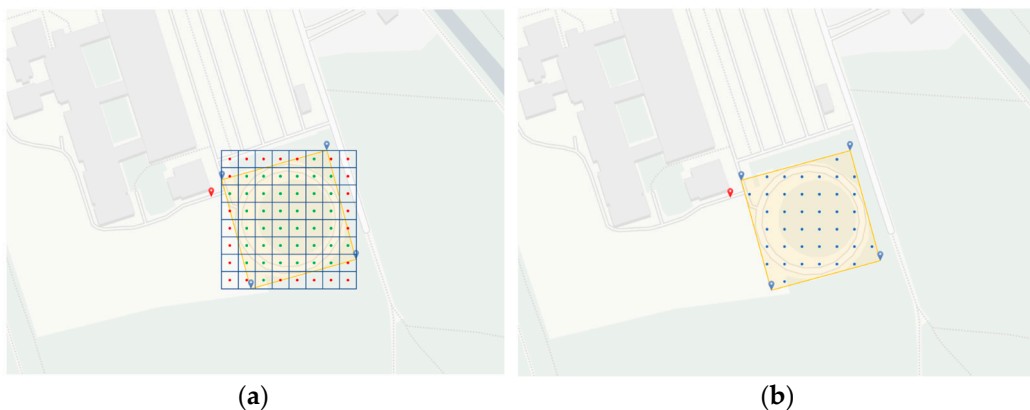

(**a**)                                                                       (**b**)

**Figure 8.** Determination of the flight path points within the bounding box (**a**) Definition of the individual points; (**b**) Defined points within the flight range.

Figure 9 shows the final flight path based on the previously described data. The starting point is highlighted with a green marker. The flight path is shown as a red line. The drone starts at its starting point, hovers there briefly, and then flies to the starting point. Once there, the defined path would be flown gradually from point to point.

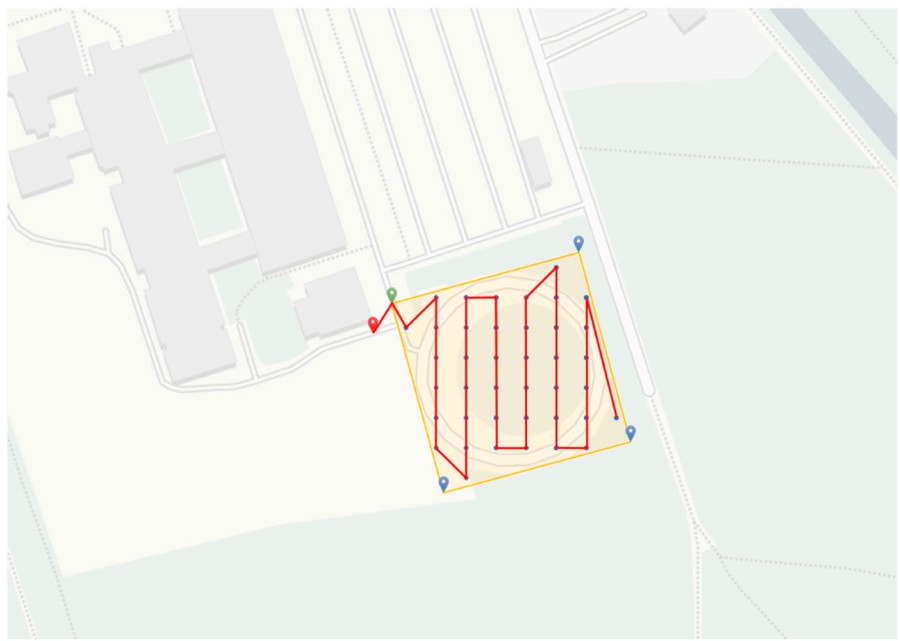

**Figure 9.** Final flight path in defined polygon, marked in red.

A coverage path planning algorithm must be implemented to plan the flight path. Since this part should not be the main focus of this work, a simple and trivial full coverage path planning algorithm was implemented. For greater efficiency, a modification of the algorithms of, for example, Refs. [38–40] should be implemented.

The implemented algorithm of this work for planning a path with complete coverage of all flight route points goes through the following process steps:

1. The corner point of the polygon closest to the position of the drone is declared as the starting point and added to the flight path.
2. All flight path points are assigned the status "unseen".
3. As long as points with this status exist, the following steps are repeated continuously:

- First, the calculation of the distance between the current position point and all unseen flight route points takes place. The Haversine distance [41] is used to calculate the distance.
- Next, the point with the shortest distance is determined as the next current situation point. However, points in unfavorable situations can be ignored and lead to a duplication of the flight route. Using a threshold, these points are included in the decision of the next attitude point.
- The new situation point is removed from the set of "unseen" points and added to the flight route.

4. If there are no further points with the status "unseen", it can be assumed that there is full coverage, and the planning process of the flight path can be completed.

Detector

The YOLOv5 detector presented in Section 2.1.2 was chosen for the detection of objects. The core of the work is based on the possibility of using the system in real time. This requires a fast yet reliable detector. YOLOv5 offers this possibility with the smallest YOLOv5 nano-model. It also provides an intuitive and user-friendly platform to enable customization of the models to the user's own needs.

*3.3. Tracker with Its Own Matching Cascade*

The DeepSORT algorithm was used as the tracker. This provides a stable baseline for simple tracking tasks. However, DeepSORT was not designed for objects to leave the scene for a longer time or to reappear at a later time by moving the camera to a different location. For this reason, some adjustments had to be made and the matching cascade for object recognition had to be extended. As a baseline, the slightly optimized DeepSORT algorithm from [42] was adopted. This has already been adapted for use in real-time applications. The implementation of the original DeepSORT still includes an offline processing style for feature extraction and would therefore be unsuitable for the use case of this work.

*Feature extraction*: In the first step of adapting the DeepSORT implementation, the feature extractor was adapted. The *MobileNetv2* neural network was used for feature extraction. This is a CNN. For this purpose, a brief description of the concepts of these neural networks was given in Section 2.1. The *MobileNetv2* was presented on the dataset of *ImageNet*. However, the focus of this work is only on two classes. Therefore, another training run was performed to fine-tune the neural network for the new classes. First, a dataset had to be created for training the network. For the person class, the same dataset as for training the YOLO network could be used with a small adjustment. For the teddy bear class, various datasets were gathered. These included the datasets from *ImageNet*, COCO, and the self-generated dataset from Section 3.2. From ImageNet and COCO, only the data that included teddy bears was retained. For both classes, only the sections of the corresponding objects were needed. For this, the ground truth labels were used to cut out the objects from each image and store them separately. Afterward, the *MobileNetv2* could be trained as a classifier. To be able to use the neural network as a feature extractor, the last layer for classification was removed.

## 4. Evaluation

The system runs the Linux Ubuntu operating system. It has an Intel Core i5 11400H processor and a GeForce RTX™ 3050 Ti laptop GPU. The processor was used for computing non-demanding problems. The graphics card was especially used for computationally intensive calculations such as neural networks.

*4.1. Evaluation Simulation Environment*

The modular structure of our software and systems allows us to test modules individually, which is helpful for ablation studies with learning algorithms and web server components. Nonetheless, system integration and rapid development are crucial, so testing the overall system and interfaces is undertaken in a system simulation. We employ the

Sphinx simulator, based on modern graphics engines (Unreal engine) and physics plug-ins (Gazebo open-source physics), to simulate the drone's flight behavior under varied environments. Photorealism allows evaluation of the computer vision component, and the freely selectable flight area is used to improve path planning and web client interfaces.

All algorithms and the overall systems were built up and tested in a simulation environment, showing promising results. Some details for the object detectors and computer vision part can be found in Table A3 in the Appendix A.

### 4.2. Real World Evaluation of the Overall System

To test the overall system and to emulate the first described scenario for non-moving objects from Section 3.1 as best as possible, the teddy bears were positioned within a staked area. The drone was to fly over the area several times in a square pattern. The following questions were to be answered:

- Could all objects be detected correctly or were there more false detections?
- Did identity switches occur within a scene?
- Did identity switches occur during the second flight, or could the objects be assigned correctly again?
- How precise was the determination of the GPS coordinates?
- Can the system process the data in real time? How many frames per second can be processed?

Figure 10 shows the staked area and the position of the placed teddy bears on a map. For an improved view, simple street maps and no satellite images were used. The three teddy bears were always placed in the same location. From the drone's starting position, two overflights were performed in a square pattern. In this context, the second flyover was used to check the system for unintentional identity changes of the tracked objects.

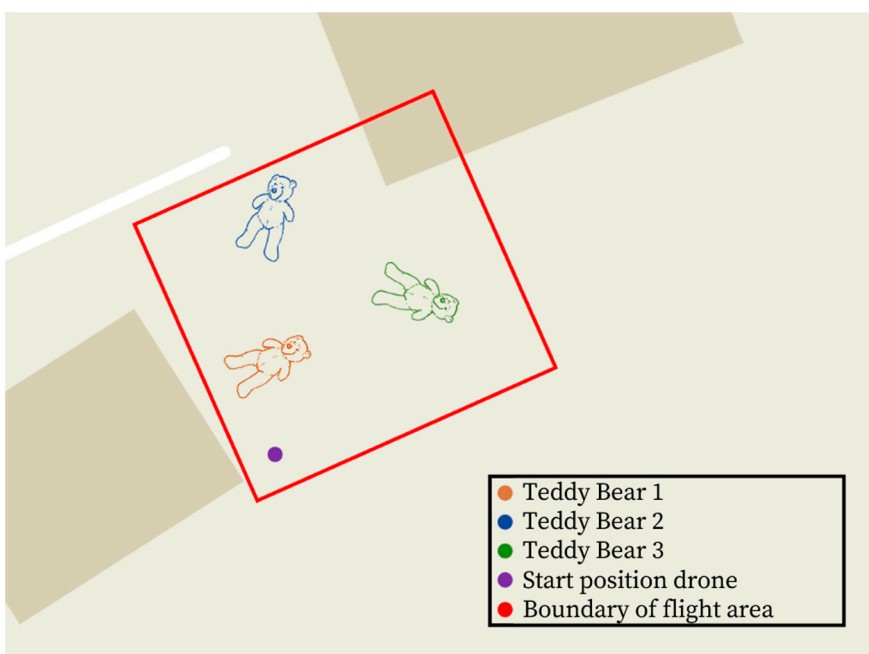

**Figure 10.** A defined area for simulating a search-and-rescue scenario.

*Evaluation of the detector*: For the evaluation of the detector, the evaluation scripts from Ultralytics could be used. These work with the metrics Precision, Recall, and $F_1$-Score. The formula of Precision is given in (3) and the formula of Recall in (4) [43].

$$Precision = \frac{TP}{TP + FP}. \tag{3}$$

*IP* (true positives) stands for the number of correct positive predictions and *FP* (false positives) for the false positive predictions. Precision indicates how many of the positive predictions are actually correct.

$$Recall = \frac{TP}{TP + FN}.$$  (4)

Here, *IP* also stands for the number of correct positive predictions, and *FN* (false negatives) for the number of false negative predictions. Recall indicates how many of the actual positive cases were correctly identified. Precision and Recall are directly dependent on each other. An improvement in Precision can lead to a decrease in Recall and vice versa. High Precision indicates accurate predictions of the model. Very few false positive predictions are made. A high Recall means that the model can detect the positive cases very well, while the number of false negative predictions is minimized.

$$F_1 = 2\frac{P \cdot R}{P + R}.$$  (5)

*P* stands for *Precision* and *R* for *Recall*. The $F_1$-score is a weighted average of Precision and Recall. It has a value between zero and one, where zero is the worst and one the best possible value. The $F_1$-Score allows the measurement of the classification accuracy for a certain class.

When training the model, a third class was also added. The watering can class was included for testing purposes only and need not be considered further in the evaluation. Figure 11 shows the *Confusion Matrix* of the YOLOv5 model used. From this, it can be deduced that the model can detect teddy bears very well. These were always detected as teddy bears only and no false detections occurred. The model seems to have slight problems with the detection of people. These could only be detected correctly 85% of the time. Otherwise, parts of the background were falsely detected as people. This increases the number of wrongly detected people. Among other things, the $F_1$-score can be derived from a confusion matrix.

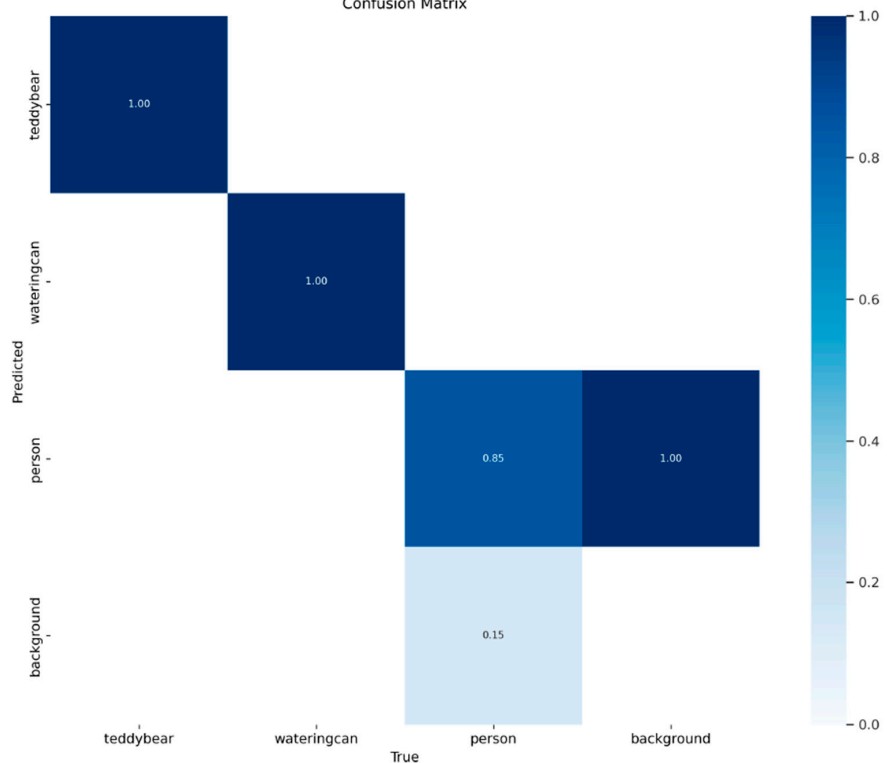

**Figure 11.** Confusion matrix of the trained model.

The $F_1$-*Score* curve is shown in Figure 12. In the figure, the $F_1$-*Score* of each class is shown individually, as well as a joint $F_1$-*Score*. Again, it can be deduced that both *Precision* and *Recall* of the *person* class are not as high as those of the *teddy bear* and watering can classes. Overall, there was a slight worsening of the $F_1$-score for the *person* class.

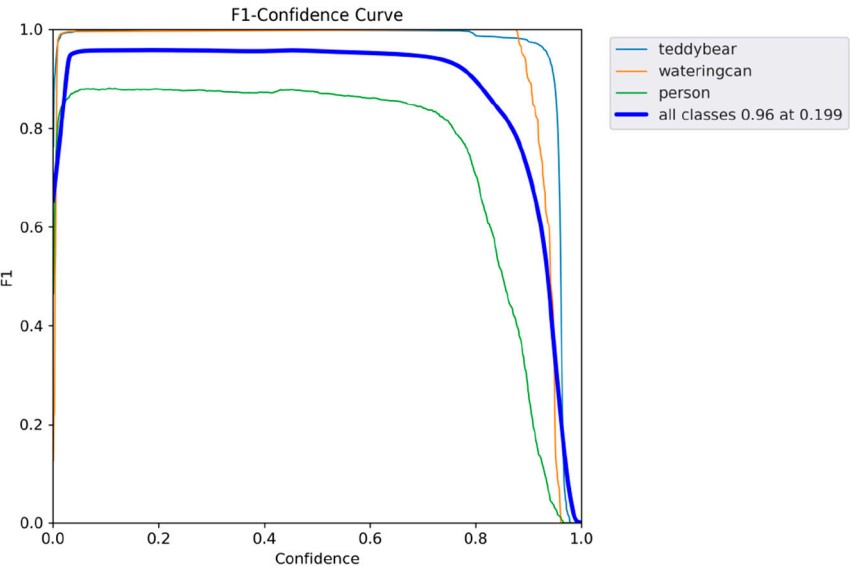

**Figure 12.** $F_1$-score curve of the trained YOLOv5 model.

The highest $F_1$ value of 0.96 is obtained with confidence of 0.199. However, it should be noted that the model may not always make the correct predictions with low confidence. Accordingly, the probability is higher that the model can be wrong in some cases. Since the $F_1$ value does not show any significant decrease up to a confidence of about 80%, it is certainly possible to consider values in this range for prediction purposes.

Figure 13 shows other metrics that can be used to classify the model. The graphs of *Precision* and *Recall* reflect the findings of the $F_1$ curve. Both values remain constant over the period of training. It can be concluded that the model has both high *Precision* and a high *Recall*. The *Validation Loss* shows an increasing trend during training but fluctuates in a range between 0.023 and 0.028, which could indicate a slight tendency to overfit the model. This could mean that the model no longer generalizes on new independent data. Since the *mean Average Precision* (*mAP*) does not show significant changes from the fifth epoch, the weights of the model should be chosen before the increasing trend of the loss. Therefore, the optimal weights of the model are in the range between the 10th and 15th epochs.

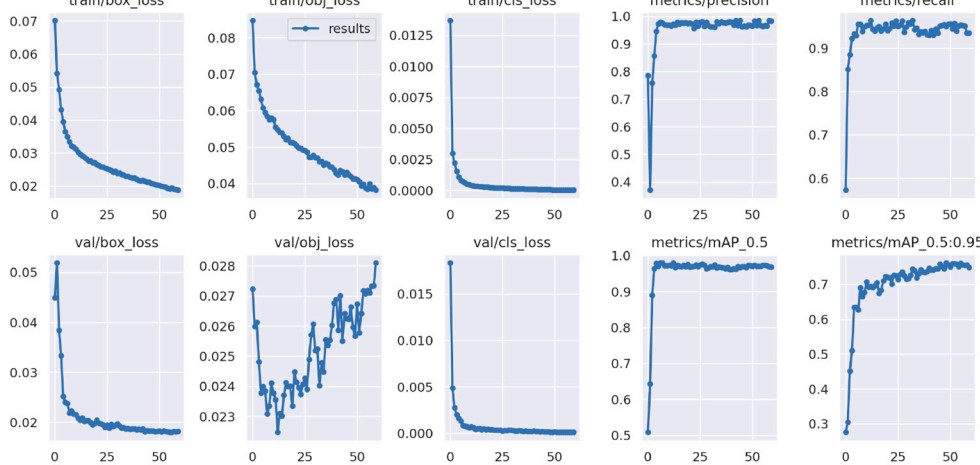

**Figure 13.** Overview of different metrics for the trained YOLOv5 model.

However, when using the trained detector, it is important to note that this is the smallest model in the YOLOv5 series. The use of a larger model could already result in a significant improvement in the predictions. However, this step could lead to losses in the speed of the system. In addition, reprocessing the dataset could go a long way toward improving the performance of the model. In creating the dataset, images from only one location were used. The times and weather conditions also remained unchanged when the dataset was created. This may cause problems with generalization to different times of day or year. Varying the shooting locations and lighting conditions could therefore enhance the quality of the dataset and provide for higher generalization to new data.

*Identity Switches*: Identity switches are cases where the tracking algorithm incorrectly assigns the identity of an object. Either the object is incorrectly assigned the identity of another object, or the track of the object is lost and the object is assigned a new identity. In both cases, it can lead to errors in tracking and identification of objects. The metric "*Multi-Object Tracking Accuracy (MOTA)*" [44] seen in Formula (6) was used to evaluate the tracking algorithm.

$$MOTA = 1 - \frac{\sum_t (m_t + fp_t + mme_t)}{\sum_t gt_t}. \tag{6}$$

where $m_t$ is the number of missed objects, $fp_t$ is the number of false positives, $mme_t$ is the number of mismatches, and $gt_t$ is the actual number of objects for time t. Usually, a concatenated sequence of frames is used for time. In the context of this work, the system should be able to recognize and correctly assign objects over longer periods. Therefore, the metric was modified, and the final results of the individual test runs were used. An overview of the runs is given in Table A1. The resulting results are listed in Table A2. By inserting the ratios from the runs into Formula (6) defined above, the following result is obtained:

$$MOTA_{Teddy} = 0.8947. \tag{7}$$

In addition to the three teddy bears, one person walked through the scene as a test. The test data show that this class was frequently detected incorrectly or not detected at all. However, it should be mentioned that the *person* class did not have to be recognized after a longer time. The persons should only be detected within a scene. The MOTA metric used also depends strongly on the detector. The tracking algorithm cannot compensate for the case where the detector does not detect an object or detects it incorrectly. The result of the test runs shows that the system works stably for non-moving objects, that the tracking algorithm detected them again on a second pass, and that there were few *Identity switches* within this class. This is probably primarily due to the inclusion of GPS coordinates. In one of the tests, two objects were assigned the same tracks until the end of the pass. This indicates a presumed *Identity switch*. In this case, the characteristics of one object were identical to the other. Therefore, the Id was assigned twice to a different teddy bear each time and was not corrected during the run.

Test runs with the simulation looked similar using the above conditions concerning non-moving objects. There, the modified MOTA metric would be close to 1. In the simulation, false detections or *Identity switches* occurred only in very rare cases. This speaks for the findings regarding non-moving target objects from the experiments with the physical drone. The system is very stable using GPS coordinates. The simulation results are listed in Table A3 in Appendix A. Unlike under real conditions, tracks within a scene were lost briefly in the simulation but could later be reassigned correctly. However, since this does not affect the performance of the overall system for non-moving objects, it is considered negligible.

### 4.3. Detecting Changes to Objects Using Cosine Similarity

For the second scenario described, live animals would be needed. Since these were not available in the context of the work, the teddy bears should simulate the moving objects. However, teddy bears are non-movable objects and look identical. For this reason, we wanted to check whether minimal changes to the objects would make a difference in distin-

guishing them. For this purpose, the features of the objects were extracted using a feature extractor. The differences in these features were determined using the *Cosine Similarity*. *Cosine Similarity* is used to measure an angle between two vectors [45]. Consequently, it indicates how similar the orientation of the two vectors is. The *Cosine Similarity* can take values from −1 to 1, where a value of −1 means that the vectors point in the opposite direction and thus have no similarities in common at all. A value of 1, on the other hand, means that the vectors point in exactly the same direction and therefore show a complete similarity. The Cosine Similarity is described with Formula (8).

$$similarity = \cos\theta = \frac{A \cdot B}{\|A\|\|B\|} = \frac{\sum_{i=1}^{n} A_i \cdot B_i}{\sqrt{\sum_{i=1}^{n} A_i^2} \cdot \sqrt{\sum_{i=1}^{n} B_i^2}}. \tag{8}$$

Here, · represents the scalar product, $\|\cdot\|$ represents the length of a vector, and *A* and *B* are the two vectors.

For the evaluation of this task, two identical images of the teddy bears were compared. On one of the images, a random number of black dots of different sizes was generated. Figure 14 shows the comparison between the original image of the teddy bear and a processed teddy bear. This allowed the differences due to the coverages to the original image to be determined as a percentage. The percentage coverage in this example is approximately 10.2%. Using the feature extractor, the features of the two images were extracted. The extracted vectors were then compared using Cosine Similarity. The number of points varied from 1 to 15. For a statistically meaningful data analysis, the process was repeated a total of 1000 times for each number of points generated to determine the average value. The results were categorized based on percentage coverage. The average per percentage point was then calculated.

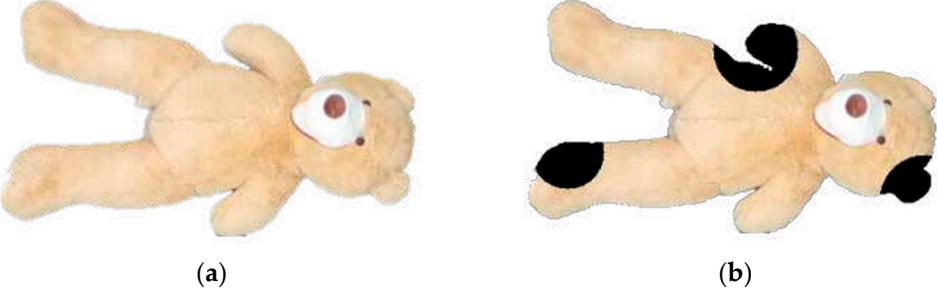

(**a**)           (**b**)

**Figure 14.** Comparison between an original teddy bear and an edited teddy bear: (**a**) Original teddy bear; (**b**) Modified teddy bear with randomly generated points. Coverage: ~10.2%.

Figure 15 shows these results in a scatter plot. The graphical representation is produced by plotting the coverage in % on the Y-axis and the average Cosine Similarity on the Y-axis. The color of the dots indicates how often percent coverage occurs. From the figure, there is a negative correlation between the coverage in % and the average *Cosine Similarity*. Increasing coverage of the teddy bear leads to a potential reduction in the similarities between the vectors representing the teddy bears. It can be concluded that discrimination or recognition of objects based on their extracted features using Cosine Similarity is quite possible. However, it must be considered that in this experiment only a partial color change of the object is involved. The resulting Cosine Similarity might not be sufficiently significant to allow a clear distinction between the objects. If Cosine Similarity is to be used to verify the identity of objects, it should also be noted that external factors such as momentary changes in lighting conditions can also have a significant effect on the result of the calculation. To simulate the changed lighting conditions, the brightness of the image was changed.

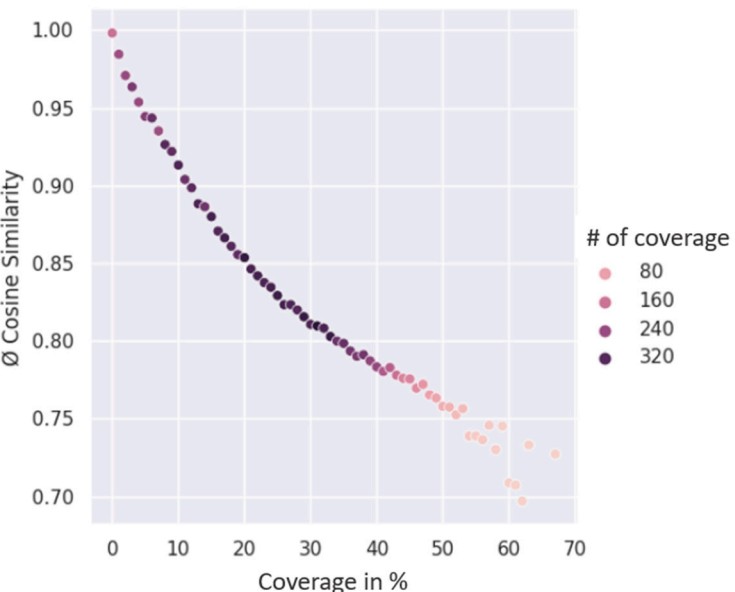

**Figure 15.** Results of the Cosine Similarity calculations.

Figure 16 shows the influence of brightness on Cosine Similarity. The brightness of the image was adjusted with values from −127 to 127. The respective similarity with the original image can be seen in the plot. From this, it can be concluded that a change in the lighting conditions has a real influence on the calculation of Cosine Similarity.

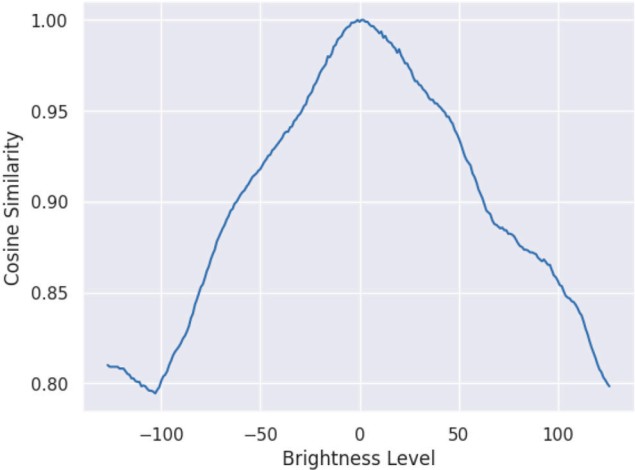

**Figure 16.** Influence of image brightness on Cosine Similarity.

Due to the inaccuracies of the bounding boxes, backgrounds could also have an influence on the discrimination of objects. To test this hypothesis, the background of the original teddy bear image was replaced with a random color. Then the features were extracted and the Cosine Similarity was calculated. This process was repeated 1000 times. The average of the calculated cosine similarities was ~0.86. Again, it is clear that Cosine Similarity can be easily influenced by changing the background.

Figure 17 shows a combination of the different manipulations of the original image. This process was also performed a total of 1000 times, with an average Cosine Similarity of approximately 0.79. Combining all the manipulations suggests that discrimination using Cosine Similarity is limited. However, only color changes have been made to the object so far. Adding altered positions or different sizes and shapes to the analysis could possibly contribute to better discrimination using *Cosine Similarity*.

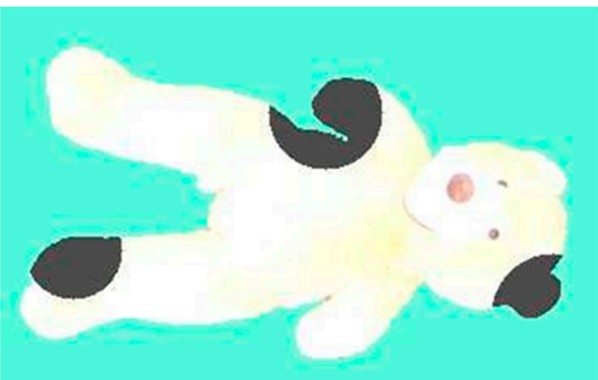

**Figure 17.** Combination of brightness, generated points, and background change.

### 4.4. Object Detection with Minimal Data Input

The goal is to provide the detector with as little training data as possible while still providing generalization.

The drone itself can be used to capture the images. For labeling the captured images, a local annotation tool is recommended. For this, the tool cvat could be used in the work. This offers the possibility to work in a local environment using a Docker container. This ensures fast, uncomplicated, and direct access.

Overall, the three models YOLOv5n, YOLOv5s, and YOLOv5m were tested for detection using minimal data overhead. The focus of this evaluation is mainly on the small nano and small models. The results of the evaluation of real-time processing from Section 4.1 showed that faster networks are necessary for real-time processing. To test minimum data overhead, the number of images used and the number of epochs trained were systematically increased. The evaluation took place using mAP. A dataset of at least 1500 images with over 10,000 labeled instances per class is advised by Ultralytics [46]. Additionally, the dataset should include images where only the background is visible. This can reduce false positive predictions. Divergent variation of images is recommended for the generalization of the model. Table 1 lists the hyperparameters used.

**Table 1.** Overview of the hyperparameters.

| Number of Training Data | 5 | 10 | 15 | 20 | 50 | 75 | 100 | 150 |
|---|---|---|---|---|---|---|---|---|
| Epochs | | 10 | 50 | 250 | 500 | | | | |
| Batch Size | | 2 | 4 | 8 | 16 | 32 | | | |

For training the models, 5 to 500 images with two instances each were used. The epochs gradually increased from 10 to 500. The batch size had to be adjusted depending on the number of images and was not freely chosen. To check the generalization ability, only images with a specific background were used for training. Successful generalization would require the models to be able to recognize objects with a different background following training. The training results are shown in Appendix A in Tables A4 and A5. It is clear that there is an increase in mAP both as the number of training data increases and as the epochs increase. This result is consistent with the expectations for this experiment. Since a precise evaluation of the detector is not possible based on mAP alone, the results had to be evaluated separately.

Precision and Recall could provide additional information about the performance of the detectors. For this purpose, the best values of both metrics were determined for each trained detector. Figure 18 shows the determined values of Precision and Recall in a scatter plot.

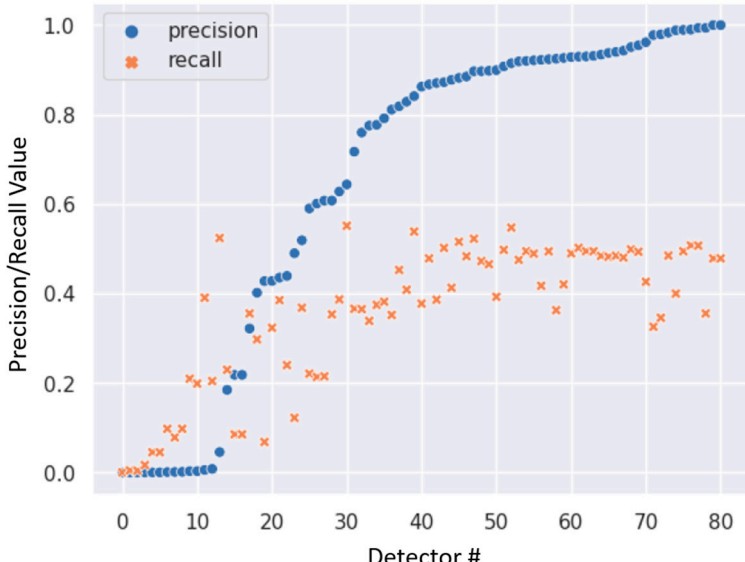

**Figure 18.** Best Precision/Recall values of all trained detectors.

From this, it can be seen that the models can identify a significant number of relevant results due to the high Precision but have difficulty in capturing the relevant results due to the mediocre Recall. This suggests that the datasets are not sufficiently representative. Therefore, a generalization does not seem to be possible. However, within the scope of this work, it is possible to train and use a detector for a specific use case in an ad hoc manner. It should be noted, however, that the detector is then trained exclusively for that specific use case and may not function effectively in another domain. For example, a change of location could cause problems due to a non-uniform subsurface.

### 4.5. Discussion of Deep Learning Algorithms Results

The evaluation of the deep learning algorithms showed important results for the system. The presented scenarios are chosen concerning the application of the drone for real-world problems. As the focus is additionally set upon a low-cost solution with minimal sensor configuration, the algorithmic choices were distinct: the detector was chosen as YOLOv5 nano, which proved to be by far the most efficient version of all one-shot detectors. It demonstrated a solid performance based on a minimal set of training data. The overall robustness can be enhanced by collecting more training data based on varying environmental influences. The incorporation of specifically trained networks for given seasons and areas (e.g., densely vegetated or desert) should further increase the accuracy.

The second deep-learning-based module is the chosen tracker. DeepSORT proved the most efficient system concerning computation time and accuracy output. The original tracker from [42] was modified and augmented to operate in real time but for significantly longer periods. The system was evaluated for re-finding livestock and the minimal changes necessary between individuals. For the given scenario with full coverage of the area in low-velocity scenarios, the performance is excellent, showing reliable tracking results for changes of up to 20% in the object area. In a detailed evaluation, the influences of minimal changes in objects on the Cosine Similarity were examined. It was found that the changes in the objects can have only partial effects on the result.

We implemented a low-cost approach for the given scenarios and achieved excellent results given the constraint of low-speed maneuvering and mediocre flight height. A comparison with commercially available products is purely qualitative, as the described scenarios are not available in basic products. More robust drones, e.g., the DJI Mavic or Agras family, in principle capable of the task, are prohibitively more expensive, up to a factor of 20 times. Drones with onboard capabilities are also too expensive and still have the bottleneck to provide the user with sufficient information and control images.

All commercial drones with enhanced camera capabilities, i.e., 8k resolution and optics, compared to the wide-angle lens and 4k resolution of our system, do not have autonomous algorithms and are priced within a range of 10 to 30 times more than our system.

## 5. Conclusions and Future Work

In this work, the design and implementation of a system for the detection and positioning of objects using a drone were presented. Using the developed system, a drone should be able to fly autonomously over a defined area, detecting various objects in real time and determining their positions. All this is subject to using only low-cost components, i.e., an affordable drone in combination with low computational power.

The work included an investigation and evaluation of different deep-learning algorithms for the detection and tracking of objects. In further exploration, the processing of video data from the drone was investigated. Subsequently, an algorithm for determining the exact GPS positions was analyzed. Details are given elsewhere [47]. Based on the results, the system was designed and subsequently implemented.

During the implementation of the system, satellite images are used to delimit the flight areas.

The system was evaluated in its entirety using a specially developed test scenario. The performance of the system was demonstrated by integrating all components in a prototype setup. This is capable of detecting static objects in real time and determining their positions with a high degree of precision. Furthermore, by extracting the features of the detections and the GPS positions, the objects can be re-identified after a longer period. A thorough study of necessary changes between the objects was conducted using the cosine distance to quantify the influences of changing conditions and objects on the system performance.

Overall, the work showed that the implementation of a real-time application for the detection and positioning of objects is possible. The system can be used for educational purposes at universities for demonstrations in the future. With an improvement in the calculation of the flight path and implementation of obstacle detection, the system could also be used in more challenging real-world scenarios.

The overall system is designed so that the replacement of the drone would require minimal effort. This would allow a more powerful drone to be used, which could execute the required neural networks using an onboard computer or cope with more heavy weather influences. Employing drones with more capable computation power would resolve the current limitation of transmitting the video data to an external computer, but also decrease the usability of the system as a low-cost competitor.

Thus, one has to choose between easy market entrance via a cost advantage or more complex scenarios with the risk of having more competition.

The directions of future work could, in principle, focus on each subsystem individually. During this work, YOLOv8 was introduced and released. This could lead to both an increase in the performance of the detection system and an increase in data throughput. Using additional training data, including diverse scenarios for weather and lighting, is another way of increasing system performance with the cost of more resources spent for acquisition.

If a more expensive chassis is chosen, additional and better sensors can be equipped and the operation area and scenarios widened significantly, but again at risk to the low-cost approach.

One important focus of future work is the design of new modules based on machine learning algorithms to showcase augmented capabilities, e.g., the assessment of rural areas for crop yield or potential hazard analysis.

**Author Contributions:** Conceptualization, S.H. and M.B.M.; methodology, S.H. and R.P.; software, R.P.; validation, S.H; investigation, S.H., M.B.M. and R.P.; resources, S.H.; writing—original draft preparation, S.H. and R.P.; writing—review and editing, S.H. and M.B.M.; visualization, S.H. and R.P.; funding acquisition, M.B.M. All authors have read and agreed to the published version of the manuscript.

**Funding:** This research is supported by the Bulgarian National Science Fund in the scope of the project "Exploration of the application of statistics and machine learning in electronics" under contract number КП-06-Н42/1.

**Institutional Review Board Statement:** Not applicable.

**Informed Consent Statement:** Not applicable.

**Data Availability Statement:** Data are contained within the article.

**Acknowledgments:** The authors would like to thank the Research and Development Sector of the Technical University of Sofia for its financial support.

**Conflicts of Interest:** The authors declare no conflict of interest.

## Appendix A

**Table A1.** Overview of the runs for testing the entire system.

| | | Ground Truth | | Detected | |
| --- | --- | --- | --- | --- | --- |
| | Flight Duration | Teddy Bears | Person | Teddy Bears | Person |
| 1. | 00:02:43 | 3 | 1 | 5 | 5 |
| 2. | 00:01:23 | 3 | 1 | 3 | 3 |
| 3. | 00:00:45 | 3 | 0 | 4 | 1 |
| 4. | 00:01:15 | 3 | 0 | 3 | 0 |
| 5. | 00:03:12 | 3 | 1 | 3 | 2 |
| 6. | 00:01:32 | 2 | 0 | 2 | 0 |
| 7. | 00:02:15 | 3 | 1 | 3 | 0 |
| 8. | 00:01:53 | 3 | 0 | 3 | 1 |
| 9. | 00:01:06 | 3 | 1 | 3 | 1 |
| 10. | 00:02:25 | 3 | 0 | 3 | 0 |
| 11. | 00:03:41 | 3 | 0 | 2 | 1 |
| 12. | 00:05:32 | 3 | 1 | 3 | 1 |
| 13. | 00:04:12 | 3 | 0 | 3 | 0 |
| Total | | 38 | 6 | 40 | 15 |

**Table A2.** Results of the runs for testing the entire system.

| | True Positives | | False Positives | | False Negatives | |
| --- | --- | --- | --- | --- | --- | --- |
| | Teddy Bears | Person | Teddy Bears | Person | Teddy Bears | Person |
| 1. | 3 | 1 | 2 | 4 | 0 | 0 |
| 2. | 3 | 1 | 0 | 2 | 0 | 0 |
| 3. | 3 | 0 | 1 | 1 | 0 | 0 |
| 4. | 3 | 0 | 0 | 0 | 0 | 0 |
| 5. | 3 | 1 | 0 | 1 | 0 | 0 |
| 6. | 2 | 0 | 0 | 0 | 0 | 0 |
| 7. | 3 | 0 | 0 | 0 | 0 | 1 |
| 8. | 3 | 0 | 0 | 1 | 0 | 0 |
| 9. | 3 | 1 | 0 | 0 | 0 | 0 |
| 10. | 3 | 0 | 0 | 0 | 0 | 0 |
| 11. | 2 | 0 | 0 | 1 | 1 | 0 |
| 12. | 3 | 0 | 0 | 1 | 0 | 0 |
| 13. | 3 | 0 | 0 | 0 | 0 | 0 |
| Total: | 37 | 4 | 3 | 11 | 1 | 1 |

**Table A3.** Results from the simulation.

|  | Ground-Truth Objects | Detected Objects | True Positives | False Positives | False Negatives |
|---|---|---|---|---|---|
| 1. | 3 | 3 | 3 | 0 | 0 |
| 2. | 3 | 3 | 3 | 0 | 0 |
| 3. | 3 | 3 | 3 | 0 | 0 |
| 4. | 3 | 3 | 3 | 0 | 0 |
| 5. | 3 | 3 | 3 | 0 | 0 |
| Total: | 15 | 15 | 15 | 0 | 0 |

**Table A4.** *mAP*—results of training with minimal data 1.

| YOLOv5n | 5 | 10 | 15 | 20 | 50 |
|---|---|---|---|---|---|
| 10 | 0.000042 | 0.00019 | 0.025 | 0.00056 | 0.025 |
| 50 | 0.0049 | 0.034 | 0.058 | 0.281 | 0.395 |
| 250 | 0.396 | 0.4065 | 0.388 | 0.407 | 0.608 |
| 500 | 0.2886 | 0.47191 | 0.403 | 0.535 | 0.679 |
| **YOLOv5s** | | | | | |
| 10 | 0.000099 | 0.00084 | 0.00084 | 0.00288 | 0.245 |
| 50 | 0.045 | 0.17 | 0.107 | 0.119 | 0.4555 |
| 250 | 0.4597 | 0.4886 | 0.495 | 0.475 | 0.61347 |
| 500 | 0.41 | 0.5147 | 0.474 | 0.676 | 0.4955 |
| **YOLOv5m** | | | | | |
| 10 | 0.00195 | 0.00145 | 0.00779 | 0.00433 | 0.0673 |

**Table A5.** *mAP*—results of training with minimal data 2.

|  | Number of Images | | | |
|---|---|---|---|---|
| YOLOv5n | 75 | 100 | 150 | 500 |
| 10 | 0.0799 | 0.1657 | 0.386 | 0.513 |
| 50 | 0.352 | 0.435 | 0.528 | 0.572 |
| 250 | 0.428 | 0.454 | 0.548 | 0.689 |
| 500 | 0.6955 | 0.375 | 0.526 | 0.58196 |
| YOLOv5s | | | | |
| 10 | 0.2799 | 0.37027 | 0.411 | 0.549 |
| 50 | 0.44449 | 0.455 | 0.52977 | 0.55443 |
| 250 | 0.5386 | 0.50692 | 0.5668 | 0.58534 |
| 500 | 0.561 | 0.51969 | 0.55095 | 0.55583 |
| YOLOv5m | | | | |
| 10 | 0.473 | 0.487 | 0.527 | 0.54845 |

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
