# Peer review of "Design and Implementation of a Camera-Based Tracking System for MAV Using Deep Learning Algorithms"

_computation, doi:10.3390/computation11120244_

Round 1
Reviewer 1 Report
Comments and Suggestions for Authors
- The introduction and the description of approach (and logic for selection of algorithms) are superb and "design" part of the article could be clearly understood.
- However, "implementation" part is relatively weaker than the "design" part raising questions and issues: (1) missing clear explanation of "simulation" and detailed method for it such as sim-environment, expected gaps between simulation and real implementation, etc. (2) main results provided do not prove any significance of the approach because it is the only instance of experiment missing any comparative studies. Authors claim that the proposed approach enables low-cost solution for detection and tracking contrast to expensive solutions that are already available. This reviewer understood that this drove the choices of more affordable detection and tracking algorithms in the context of performance-cost balance. Without direct or even indirect comparison of performance between the proposed solution and the existing solutions, this reviewer was unable to access the significance of this report.
Author Response
Editorial Board of Computation
Respected Reviewers
Response letter
Thank you for reviewing the manuscript, and for the insightful comments from the editor and reviewers that helped us improve its quality.
Please find our response below (all changes we made are highlighted in the manuscript to facilitate the revision process).
Reviewer 1
Thank you very much for reviewing the work.
Thank you for your feedback and the positive impressions given for the design part.
Concerning your regards on the implementation part, we added details regarding the simulation, especially in Section 4.1. As you assumed, there are some gaps between the real world and the simulation environment, but they showed to be neglectable in comparison to the time spared for development.
The second part of your proposal is harder to fulfill, as the commercially available systems do not lay open their tracking algorithms and performance. Therefore, one always has to use subjective quality assessment for the results. What can be done, is to list some prices of commercially available drones and their respective software. We added an explanation and paragraph in the restructured section “discussion of the results” 4.5 picking some commercially available drones that at least seem capable of performing our applications (alas, not in the exactly same way).
We hope, that the added information and restructuring enhance the implementation part significantly,
Greetings and kind regards.

Reviewer 2 Report
Comments and Suggestions for Authors
After careful reading, the focus of this article is on the implementation and application of the method, and there are shortcomings in the proposal of new methods. DeepSORT, YOLO and other methods are all existing methods, and the author has verified their practical applications. There are no specific features related to MAV, and the relevant methods can also be applied to other forms of tasks. Suggest the author to provide the connection, characteristics, and necessity between the processing method and practical application. In addition, the author needs to consider the stability and performance verification of the method in complex scenarios and complex flight trajectory situations.
Author Response
Editorial Board of Computation
Respected Reviewers
Response letter
Thank you for reviewing the manuscript, and for the insightful comments from the editor and reviewers that helped us improve its quality.
Please find our response below (all changes we made are highlighted in the manuscript to facilitate the revision process).
Reviewer 2
Thank you for your feedback concerning our proposed article. As you mentioned, the employed methods are from the field of deep learning, but not specifically published for MAV applications. We do not claim the methods to be new, but we adopt them for our purpose. Each method was chosen specifically for the purpose of our application and a substantial part of the work is the process and evaluation of choosing the method. For example, as an object detector, we chose YOLOv5, which is a well-established method to detect objects. For our purpose, we had to choose the most efficient implementation, compare it against R-CNN-like methods, and tweak the parameters specifically for our purpose (this resulted in a decision for “YOLOv5 nano”, against the newer YOLOv8, which was more hardware intensive), as shown exemplarily in Section 4.3 for the detection stage.
We tried to highlight the applications by introducing the scenarios, of which each requires certain functions to perform well. These functional modules are chosen to be solved by a combination of algorithms and methods showing top performance and state-of-the-art in their field. After carefully selecting the methods, they are implemented and integrated into the overall system to perform the desired tasks.
For your second remark, we restructured section 4, adding a subsection to discuss the results and findings of the real-world situations in Section 4.5. We do not regard more complex scenarios, as we intend to solve the two considered cases, which are becoming more complex purely by enlarging the application, e.g. tracking more animals than the three exemplarily selected bears. You have a valid point to question the system performance for more complex trajectories, but for this purpose, we use the indicated hardware, i.e. the Parrot Anafi MAV. This drone is stabilized and limited in its flight speed, which means, that no complex maneuvers arise, due to the path control and stabilization algorithms. The camera, which is our main sensor, stays facing toward the ground due to the gimbal, and the flight patterns are chosen according to section 3., ensuring no fast flights or narrow curves.
We hope, that the changes in structure and minor additions as well as these explanations help to alleviate the respective points.
Kind regards.

Reviewer 3 Report
Comments and Suggestions for Authors
Regarding the manuscript titled "Design and Implementation of a Camera-Based Tracking System for MAV Using Deep Learning Algorithms Convolutional Neural Networks," the author developed a camera-based tracking system for low-cost drones with the aim of real-time object detection and localization. Deep learning algorithms were employed and evaluated for system performance. The manuscript is well-written, with a well-organized methodology. However, some revisions are needed, and I would like to discuss the specific areas that require attention.
I would like to commend the authors for their valuable study, which exhibits scientific significance and a well-structured methodology. Nonetheless, minor editing is necessary prior to publication. After carefully reviewing the manuscript twice, I would like to highlight a few points that should be addressed:
The abstract should be more comprehensive, including both the method and the detailed findings. It is important to provide a careful mention of the study's outcomes in the abstract.
It would be beneficial to add a section that reviews previous studies in the field. This addition would greatly enhance the readability of your manuscript.
Given the novelty of this field, it would be advantageous to include a section titled "Suggestions for the Future," which can serve as a roadmap for future research endeavors. This addition will further enhance the manuscript's value.
Overall, the manuscript demonstrates significance, but attention to these points will enhance its quality and ensure its readiness for publication.
Comments on the Quality of English LanguageDear Editor, I am writing to provide feedback on the manuscript titled "Design and Implementation of a Camera-Based Tracking System for MAV Using Deep Learning Algorithms Convolutional Neural Networks." The author successfully developed a camera-based tracking system for affordable drones, focusing on real-time object detection and localization. Deep learning algorithms were utilized and assessed for system performance. The manuscript exhibits a well-written narrative and a logically structured methodology. However, I would like to discuss areas that require revision and attention.
Kind regards,
Author Response
Editorial Board of Computation
Respected Reviewers
Response letter
Thank you for reviewing the manuscript, and for the insightful comments from the editor and reviewers that helped us improve its quality.
Please find our response below (all changes we made are highlighted in the manuscript to facilitate the revision process).
Reviewer 3
Thank you for your positive feedback concerning our article as well as the proposed improvements.
We reformulated the abstract to incorporate your advice of adding some major results and paper outcomes.
In addition, we added an enlarged introduction section with some related work to better place the article in a research framework (please refer to L53-L85).
We also restructured the final Sections 4 and 5 to incorporate more clearly possible work for future systems and improvements to our approach.
Again, thank you very much for the positive response and suggested enhancements.
Kind regards.

Round 2
Reviewer 2 Report
Comments and Suggestions for Authors
no further comments.